# Mining Single Nucleotide Polymorphism (SNP) Markers for Accurate Genotype Identification and Diversity Analysis of Chinese Jujube (*Ziziphus jujuba* Mill.) Germplasm

**Lihua Song [1],\*, Bing Cao [1] , Yue Zhang [1] , Lyndel W. Meinhardt [2] and Dapeng Zhang [2]**

1 School of Agriculture, Ningxia University, Yinchuan 750021, China; bingcao2006@126.com (B.C.); 13722302660@163.com (Y.Z.)
2 USDA-ARS, Sustainable Perennial Crop Laboratory, Beltsville, MD 20705, USA; lyndel.meinhardt@usda.gov (L.W.M.); dapeng.zhang@usda.gov (D.Z.)
\* Correspondence: slh382@nxu.edu.cn

**Abstract:** Chinese jujube (*Ziziphus jujuba* Mill.) is an economically important fruit tree with outstanding adaptability to marginal lands and a broad range of climate conditions. There are over 800 cultivars, mostly landraces from China. However, a high rate of mislabeling in Chinese jujube germplasm restricts the sharing of information and materials among jujube researchers and hampers the use of jujube germplasm in breeding. In the present study, we developed a large panel of single nucleotide polymorphism (SNP) markers and validated 288 SNPs by genotyping 114 accessions of Chinese jujube germplasm. The validation resulted in the designation of a set of 192 polymorphic SNP markers that revealed a high rate of synonymous mislabeling in the jujube germplasm collection in Ningxia, China. A total of 17 groups of duplicates were detected, encompassing 49 of the 114 Chinese jujube cultivars. Model-based population stratification revealed two germplasm groups, and the core members of the two groups showed a significant genetic differentiation (Fst = 0.16). The results supported the hypothesis that the cultivated Chinese jujube had multiple origins and multiple regions of domestication. The Neighbor-Joining dendrogram further revealed that this collection is comprised of multiple sub-groups, each including 1-13 closely related cultivars. Parentage analysis of cultivars with known pedigree information proved the efficacy of using these SNP markers for parentage verification. A subset of 96 SNPs with high information index was selected for future downstream application including gene bank management, verification of pedigrees in breeding programs, quality control for propagation of planting materials and support of the traceability and authentication of jujube products.

**Keywords:** genetic integrity; genetic redundancy; gene bank; mislabeling; DNA fingerprinting; molecular markers; domestication; off-type; Chinese date

## 1. Introduction

Chinese jujube (*Ziziphus jujuba* Mill.) is a diploid fruit crop (2n = 2x = 24) in the Rhamnaceae family. This plant species is native to China, with its putative center of origin located in the Yellow River basin [1–3]. Chinese jujube (hereafter referred to as jujube) is a multipurpose tree cultivated for fruits and has a tremendous economic importance. It is one of the earliest domesticated fruit trees in China, with a history of utilization going back more than 7000 years [2–4]. Recent research suggests that the current cultivars of Chinese jujube were originally selected from sour jujubes (*Ziziphus jujuba* Mill. var. *spinosa*), which are still widely distributed in Northern, Central, and Southwestern China [1,3,4]. Jujube has become increasingly popular in China and abroad for its outstanding adaptability, nutritious fruits, and many attributes that are utilized in food and traditional medicine. It is an ideal economic crop for arid and semiarid areas of temperate and subtropical regions where most common fruit trees cannot be grown. Presently it ranks 7th among fruit tree

crops in China in terms of cultivation area (600,000 ha), with an annual production of more than 1.1 million metric tons. This tree crop has been introduced into about 50 countries throughout Asia, Europe, Africa, the Americas, and Oceania [2].

At least 853 jujube cultivars were maintained in various ex situ repositories in China [2]. Jujube cultivars vary widely in morphological characteristics, fruiting season, yield, disease resistances, and fruit quality. Most of these cultivars are landraces or farmer selections [1,3]. This crop has a long history of introduction and exchange among different regions in China, as well as different countries and continents. Like many other perennial crops, jujube germplasm can be maintained by vegetative propagation, and it is commonly exchanged as clones. However, the exchange of vegetative planting materials has also resulted in problems for jujube genebank curators and breeders because records and labels of the cultivars have not always followed the same naming conventions, or accessions lack information regarding their correct identity. Therefore, mislabeled genetic materials (homonyms and synonyms) are common among jujube cultivars and that restricts the sharing of information and materials among jujube researchers and hampers the use of jujube germplasm in breeding and cultivar deployment.

DNA fingerprinting has been widely applied to assist management of genetic resources and crop improvement [5]. Various types of molecular markers have been applied in jujube germplasm management, especially simple sequence repeats (SSRs) [6–9]. Using 24 pairs of SSR markers, Xu et al. (2016) analyzed genetic integrity of 962 jujube accessions and reported a high rate of mislabeling. A total of 448 accessions, which accounted for 47.6% of the jujube in the tested collection, possessed synonymous mislabeling (accessions with different names but sharing identical SSR genotypes). Such high rate of mislabeling hinders accuracy and efficiency in management and the use of jujube germplasm in breeding, both in China and abroad. The lack of correctly identified jujube germplasm has hindered the use of true-to-type parental lines, thus severely limited the development of improved jujube cultivars [9,10].

Single Nucleotide Polymorphisms (SNPs) in plant genomes are the most abundant class of naturally occurring genetic polymorphisms. Compared to SSR markers, SNP analysis does not require DNA separation by fragment length and, therefore, can be automated in high-throughput assay formats with much lower error rates and greater efficiency and reproducibility between laboratories. These advantages have made SNPs the markers of choice for accurate cultivar identification and diversity analysis, as well as for pedigree verification in breeding programs, accreditation of planting materials and seedling nurseries, and for the authentication and traceability of high-value cultivars for premium markets [7,11–13].

Recently, next-generation sequencing (NGS), especially genotyping-by-sequencing (GBS), was used for diversity analysis, genetic linkage map construction and association mapping in jujube [14–16]. While highly informative, direct use of a GBS approach for accurate cultivar identification is not practical because of the relatively high error rate in the sequences. Furthermore, GBS is not cost effective in large scale downstream applications where many jujube accessions need to be genotyped (e.g., accreditation of seed garden and plant propagation nurseries). Instead, an array-based DNA fingerprinting approach that uses a small set of highly reliable SNP markers is preferable for a broad range of research needs and field applications.

Ample genomic resources have been developed for jujube, including ESTs, DNA and transcriptome sequences, and draft genomes of cv. 'Dongzao' and 'Junzao' [17,18]. These readily available genomic resources provide opportunities to mine new SNP markers for jujube germplasm management and breeding. The objectives of the present study were to develop SNP markers through data mining of sequences available in the public domain and to apply them for jujube genebank management. The results reported herein represent the first SNP discovery and validation study in jujube, demonstrating the utility of published genomic resources as an approach for rapid development of high-quality genotyping tools.

These SNP markers, as well as the genotyping method, will be particularly useful for jujube germplasm management, breeding programs, and propagation of planting materials.

## 2. Materials and Methods

### 2.1. Discovering Jujube SNP Markers through Data Mining

SNP data mining was performed using sequence data of 36 *Ziziphus jujuba* genotypes (SRR3095649 to SRR3095689, SRR3310162 to SRR3310166, SRR5041640, SRR5041641, SRR5041644, SRR5041645), as well as the related species *Ziziphus mauritiana* (SRR6267272) and *Ziziphus spina-christi* (SRR6277366), which were deposited in the NCBI Sequence Read Archive (SRA) database. These SRA reads were downloaded from the database and mapped on the jujube reference genome (JREP00000000) [17] using the BWA program [18]. The Genome Analysis Toolkit (GATK) package v 3.5 [19] was used for SNP calling using HaplotypeCaller with default parameters. Then the hard filters (parameters: QD < 2.0 || FS > 60.0 || MQ < 40.0 || MQRankSum < −12.5 || ReadPosRankSum < −8.0) were applied to exclude low-quality alleles. Four sequencing datasets (SRR3081153, SRR3081197, SRR3081340, and SRR3081342), which were used in jujube genome-assembly, were also downloaded and included in GATK SNP calling steps. These data were used as internal references to correct error or ambiguous sequences in jujube genome assembly. Among 36 jujube genotypes, the polymorphic loci (MAF > 0.10) were selected as candidate SNP loci. To select high-quality SNPs for experimental validation, any SNPs that had other possible adjacent SNP sites 80 bp upstream or 80 bp downstream were eliminated. From the discovered putative SNPs, a subset of 288 putative SNPs was selected for validation test using the nanofluidic array genotyping system (Fluidigm Co, South San Francisco, CA, USA). The primers of the selected 288 SNPs were designed by Fluidigm and applied on the selected jujube cultivars for validation.

### 2.2. Plant Materials and DNA Extraction

A total of 114 jujube cultivars (Table 1) were used in the present study. These jujube germplasm accessions were maintained in the jujube collection in Yinchuan, Ningxia, China. For DNA extraction, three fully expanded healthy leaves were harvested and the leaves were freeze-dried. The DNeasy Plant Mini kit (Qiagen Inc., Valencia, CA, USA), was used to extract DNA from the dried jujube leaves. A TissueLyser II (Qiagen Inc.) was used to disrupt the dry leaf tissue samples with high-speed shaking (30 Hz for 1 min) using Lysing Matrix A (MP Biomedicals. Solon, OH, USA) as described in Fang et al., 2013. A NanoDrop spectrophotometer (Thermo Scientific, Wilmington, DE, USA) was used to determine DNA concentration by absorbance at 260 nm and to estimate DNA purity at ratios of 260:280 and 260:230.

**Table 1.** List of 114 Chinese jujube cultivars maintained in Ningxia, China genotyped by 288 SNP markers in the present study.

| Name of Cultivars | Source (Province) | Name of Cultivars | Source (Province) |
|---|---|---|---|
| Fuyangmutouzao | Anhui | Zaoqiuhong | Ningxia |
| Beijingbenzao | Beijing | Changtanzao | Ningxia |
| Beijingpaoapaozao | Beijing | Zhongningyuanzao | Ningxia |
| Jingzao60 | Beijing | Dalixiaodundunzao | Shaanxi |
| Shaizao | Beijing | Dieyazao | Shaanxi |
| Yingluozao | BeiJing | Dongzao | Shaanxi |
| Beibeixiaozao | Chongqing | Ganweibazao | Shaanxi |
| Gansudongzao | Gansu | Goutouzao | Shaanxi |
| Minqinxiaozao | Gansu | Jiaxianyazao | Shaanxi |
| Zunyitianzao | Guizhou | Muzao | Shaanxi |
| Cangxiantunzizao | Hebei | Pingshunbenzao | Shaanxi |
| Chuanlingzao | Hebei | Pingshunjunzao | Shaanxi |
| Fengmizao | Hebei | Qiyuexian | Shaanxi |

**Table 1.** *Cont.*

| Name of Cultivars | Source (Province) | Name of Cultivars | Source (Province) |
|---|---|---|---|
| Hebei 13# | Hebei | Zhongcaobenzao | Shaanxi |
| Jinsixiaozao | Hebei | Dalingzao | Shandong |
| Longzao | Hebei | Fucuimizao | Shandong |
| Malianxiaozao | Hebei | Laolingxiaozao | Shandong |
| Pozaozhibian 1# | Hebei | Liaochengyuanlingzao | Shandong |
| Shulutangzao | Hebei | Linqinzao | Shandong |
| Xianxianmatouzao | Hebei | Luzao 2# | Shandong |
| Zanhuangchangzao | Hebei | Luzao 5# | Shandong |
| Zaocuiwang | Hebei | Suyuanlingzao | Shandong |
| Bianhesuanzao | Henan | Tengxiantangzao | Shandong |
| Dongzao 2# | Henan | Baodeyouzao | Shanxi |
| Henanlongzao | Henan | Mamazao | Shanxi |
| Jinmangguozao | Henan | Jianjianzao | Shanxi |
| Lingbaoyuanzao | Henan | Jiaochengduanzao | Shanxi |
| Fuyangtangzao | Henan | Jingudazao | Shanxi |
| Fuyangxiaozao | Henan | Jinzao | Shanxi |
| Xinzhengjixinzao | Henan | Jiuyuehan | Shanxi |
| Hunanchangzao | Hunan | Lizao | Shanxi |
| Mifengzao | Hunan | Linfenmizao | Shanxi |
| Ruchengzao | Hunan | Linfenmugedazao | Shanxi |
| Tangtouzao | Hunan | Linyibenzao | Shanxi |
| Xupuchengtuozao | Hunan | Linyitiansuanzao | Shanxi |
| Xupumizao | Hunan | Linglingzao | Shanxi |
| Xupushatangzao | Hunan | Mopanzao | Shanxi |
| Xupushibingzao | Hunan | Puchengjinzao | Shanxi |
| Xupusuanyuanzao | Hunan | Sumuzao | Shanxi |
| Lengsizao | Jiangsu | Taigulinglingzao | Shanxi |
| Penzao 2# | Jiangsu | Taiyuanyuanzao | Shanxi |
| Dasuanzao | Ningxia | Tuanzao | Shanxi |
| Dayuanzao | Ningxia | Wanrongcuizao | Shanxi |
| Jinchang 1# | Ningxia | Xiangfenyuanzao | Shanxi |
| Lingwusuanzao | Ningxia | Yongjijidanzao | Shanxi |
| Lingwuyuanzao | Ningxia | Yucijiuyueqing | Shanxi |
| Lingwuchangzao | Ningxia | Yucituanzao | Shanxi |
| Longzhu 1# | Ningxia | Yuanquzao | Shanxi |
| Longzhu 2# | Ningxia | Yunchengjinzao | Shanxi |
| Longzhu 3# | Ningxia | Yunchengpopozao | Shanxi |
| Suanzao | Ningxia | Alaayuancuizao | Xinjiang |
| Tongxinyuanzao | Ningxia | Hamidazao | Xinjiang |
| Tongxinyuanzao | Ningxia | Huizao | Xinjiang |
| Yuanzao | Ningxia | Kunmingzao | Yunnan |
| Yuanzao 1# | Ningxia | Yiwudazao | Zhejiang |
| Yuanzao 2# | Ningxia | Yiwuezizao | Zhejiang |
| Yuanzao 3# | Ningxia | Yiwumianxuzao | Zhejiang |

*2.3. Validation of Putative SNPs*

A nanofluidic genotyping system was used to evaluate the putative SNP markers for cultivar identification. The Assay Design Group at Fluidigm Corp. (South San Francisco, CA, USA) designed and manufactured the putative SNP primers for competitive allele-specific PCR, enabling bi-allelic scoring of SNPs at specific loci (KBioscience Ltd., Hoddesdon, UK). An EP1 imager (Fluidigm Corp., South San Francisco, CA, USA) was used to acquire fluorescent images of the endpoint reactions in the 96.96 IFC and Fluidigm Genotyping Analysis Software (Fluidigm Corp., South San Francisco, CA, USA) was used to analyze the data.

*2.4. Data Analysis*

Duplicate cultivars were identified using pairwise multilocus matching among all individual samples. DNA samples that were fully matched at all genotyped SNP loci were considered the same cultivar or clones. The procedure of multilocus matches, as implemented in the program GenAlEx 6.5 [20], was used for computation. The probability of identity among siblings (PID-$_{SIB}$), which is the probability that two sibling individuals drawn at random from a population have the same multilocus genotype, was used to measure the statistical rigor of the matching result. The overall PID provides the minimum essential number of loci required to resolve all individuals and relatives in a group. After duplicate identification, the redundant samples were removed and only one genotype from each duplicate group was retained and included in consequent diversity analysis. Summary statistics, including minor allele frequency, observed heterozygosity, expected heterozygosity, and Shannon's information index were computed, using the software GenAlEx 6.5 [20].

Population structure of the jujube samples was determined using a model-based Bayesian cluster analysis software STRUCTURE v2.3.4 [21]. The admixture model was applied and the number of clusters (K-value), indicating the number of genetic clusters, was set from 1 to 10. The analyses were carried out without assuming any prior information about the genetic groups or geographic origins of the samples. Ten independent runs were assessed for each fixed number of clusters (K value), each consisting of 100,000 iterations after a burn-in of 200,000 iterations. The Delta K value [22] was used to detect the most probable number of clusters using the online program STRUCTURE HARVESTER [23]. Permutation was performed using the computer program CLUMPPv1.1.1 [24] and the resultant outputs were then visualized using computer program Distruct v1.1 [25].

Distance-based multivariate analysis was performed on the individual data. Pairwise genetic distances were computed using the Distance option, and Principal Coordinates Analysis (PCoA) within the GenAlEx 6.5 program [20]. Both distance and covariance were not standardized. In addition, a cluster analysis using the neighbor-joining (NJ) method was used to further examine the genetic relationship among the cultivars with unique SNP profiles. Nei's distance [26] was chosen as a genetic distance measurement for the individual accessions with the program MICROSATELLITE ANALYZER [27]. A dendrogram was generated from the resulting distance matrix using the NJ algorithm available in PHYLIP version 3.697 [28] and the tree was constructed with the program Fig Tree v 1.4.3 [29].

To test the efficacy of using these SNPs for pedigree verification in jujube germplasm, seven cultivars with known parents (per literature records), were selected for parentage analysis (Table 1). It is known that these seven cultivars were selected from true seedlings (in contrast to clonal selection) but their parentage was either partially known or unknown. These cultivars were considered 'offspring' for which parentage analyses were carried out using the rest of the cultivars as potential candidate parents. A likelihood-based method implemented in the program CERVUS 3.0 was used for computation [30,31]. A likelihood ratio (LOD score) was calculated for each parent–offspring pair. Critical LOD scores were determined for the assignment of parentage to a group of individuals without knowing the maternal or paternal parent. Simulations were run for 10,000 cycles, assuming that 20% of candidate parents were sampled, and a total of 95% of loci was typed with a 1% typing error rate. The most probable single mother (or father) for each offspring was identified based on the critical difference in LOD scores (D) between the most likely and the next most likely candidate parents at greater than 95% confidence [30,31].

To facilitate future-large scale application of these SNPs in jujube genotyping, a core set of 96 SNPs was selected out of the 192 SNPs. Quality Assurance Module from SNP Variation Suite version 8 (SVS8; Golden Helix Inc., Bozeman, MN, USA) was applied to remove SNPs that were in high level of linkage disequilibrium (LD) with each other ($r^2 \geq 0.5$). Then the final core set of 96 SNPs was selected based on the Shannon's information index values. The accumulated PID value was computed for these 96 SNP markers following the

method of Waits et al., (2001), using GenAlEx 6.5 [20]. Genetic distances among the jujube cultivars were computed using the selected 96 SNP markers. A Mantel test was performed between the distance matrix based on the full panel of 192 SNPs and the matrix based on the selected 96 SNPs, using the same computer program.

## 3. Result

### 3.1. Data Mining and SNP Discovery

A total of 41 files containing high-throughput sequencing data were downloaded from NCBI, accounting for 376.43 giga nucleotides. NGS QC Toolkit (v. 2.3.2) software was used to remove reads with 20% or more low quality bases (Phred score < 20) [32]. High-quality reads from all sequencing data were then compiled for alignment using the short-read mapping program BWA. SNP calling by GATK was applied for all sequencing data separately, resulting in a large number of potential SNPs. On average, around 2,500,000 potential SNPs were called in each jujube genotype. An in-house Perl script was then used to merge all potential loci, resulting in a total of 11,366,557 SNPs. To select high-quality SNPs for experimental validation, SNP sites having an adjacent SNP site either 80 bp upstream or 80 bp downstream were eliminated. In total, 32,249 putative SNPs, including coding gene regions and intergenic regions that covered all jujube chromosomes, were obtained, which were applicable for SNP experimental validation. Detailed information of these putative SNPs is presented in Supplementary Materials (Supplementary Data 1).

A total of 288 putative SNPs were selected for validation testing. Out of 288 SNP markers, only 12 failed, likely due to the sequence complexity or the presence of polymorphisms within the flanking sequences. Among the 276 successful SNPs, 40 were monomorphic across the 114 samples (i.e., only one SNP variant was identified in all individuals) or the frequency was lower than 0.02. These monomorphic markers may have resulted from errors in sequencing, which then led to the incorrect identification of SNPs. It is also possible that some of these SNPs may correspond to rare alleles that were not present in the tested set of jujube accessions. From the remaining 236 SNP markers, a total of 192 polymorphic SNPs were selected based on their no-call rate and consistency of genotyping result. Primers with a no-call rate above 5% were excluded. This final set of 192 SNPs was included in the subsequent data analysis. The flanking sequences for these 192 SNPs are listed in Supplementary Data 2, whereas the genotyping result generated by the 192 SNP markers for all 114 analyzed Chinese jujube cultivars is presented in Supplementary Data 3.

### 3.2. Cultivar Identification

SNP profiles of the multiple leaf samples from the same jujube cultivars showed that genotyping results were highly consistent, as shown by the high repeatability of internal controls (Supplementary Data 3). An example showing the multilocus SNP data among jujube cultivars was presented in Table 2. Multilocus matching of SNP fingerprints revealed a high rate of duplicates in this jujube collection. Out of the 114 analyzed cultivars, a total of 49 cultivars could be classified into 17 synonymous groups (Table 3). The number of cultivars in each synonymous group ranged from two to eight. The probability that two jujube cultivars will have the same genotype at the 192 SNP loci is approximately 1 in 1,000,000 as computed by the multilocus matching procedure found in GenAlEx 6.5 [20]. From each of the synonymous groups, only one cultivar was retained and used for subsequent diversity analysis. This procedure led to the identification of 79 genotypes that had unique SNP profiles.

Descriptive statistics were then computed for the 192 polymorphic SNPs across the 79 jujube cultivars, and the results are presented in Supplementary Data 4. The mean information index was 0.577, ranging from 0.010 to 0.693. The observed heterozygosity ranged from 0.013 to 0.842 with an average of 0.355, whereas the mean expected heterozygosity was 0.350, ranging from 0.008 to 0.500 (Supplementary Data 4).

**Table 2.** Examples of DNA fingerprints based on the array of 192 SNPs for jujube cultivar identification (showing truncated profiles).

| Cultivars | Zj002 | Zj018 | Zj019 | Zj022 | Zj025 | Zj026 | Zj028 | Zj030 | Zj031 | Zj032 | Zj033 | Zj034 | Zj035 | Zj036 | Zj037 | Zj038 | Zj039 | Zj040 | Zj041 | Zj043 |
|---|---|---|---|---|---|---|---|---|---|---|---|---|---|---|---|---|---|---|---|---|
| Alaayuancuizao | C T | T T | A A | G G | G T | G G | C T | C C | T T | C C | T T | A A | G G | G T | C T | G G | C C | T T | C C | A A |
| Baodeyouzao | C T | C T | A A | A G | G T | G G | C T | C C | T T | C C | T T | A A | G G | G T | C T | G G | C C | T T | A C | A A |
| Beibeixiaozao | C C | T T | T T | A A | G G | G G | C C | C C | T T | C C | T T | A G | G G | G G | C T | G G | C C | T T | C C | A A |
| Beijingpaoapaozao | C C | T T | A A | A A | G G | G G | C C | T T | C T | C C | T T | A G | G G | G G | C T | C G | C G | C T | A C | A T |
| Chuanlingzao | C C | T T | T T | A A | G G | G G | C C | T T | C T | C C | T T | A A | G G | G G | C C | C G | C G | C T | A C | A T |
| Dalingzao | T T | C T | A T | A G | G G | G G | T T | C C | C T | C C | T T | A A | G G | G G | C C | C G | C G | C T | A A | A T |
| Dalixiaodundunzao | T T | C T | T T | A A | G G | G T | C C | C C | T T | C C | T T | A A | G G | G G | C C | C G | C G | C T | A C | A T |
| Dasuanzao | C C | T T | T T | A A | G G | G T | C C | C C | T T | C C | C T | A A | A G | G G | C C | G G | C C | T T | A C | A A |
| Dieyazao | T T | C T | A A | G G | G G | G G | T T | T T | C T | C C | T T | A A | G G | G T | C C | C G | C G | C T | A A | A T |
| Dongzao | C C | T T | T T | A A | G T | G G | C T | C C | T T | C C | T T | A A | G G | G T | C T | G G | C C | T T | A C | A A |
| Fengmizao | T T | C T | A T | A G | G G | G G | C C | C T | T T | C C | T T | A G | G G | G G | C T | C G | C G | C T | C C | A T |
| Fuyangmutouzao | C C | T T | T T | A A | G G | G G | C T | C C | T T | C C | T T | A G | G G | G T | C T | G G | C C | T T | A C | A A |
| Gansudongzao | T T | C T | T T | A A | G G | G G | C C | C C | T T | C C | C T | A A | A G | G T | C C | C G | C G | C C | C C | A T |
| Hamidazao | C C | T T | T T | A A | G G | G T | C C | C C | T T | C C | T T | A A | G G | G G | C C | G G | C C | T T | A C | A A |
| Hebei 13# | C T | T T | A A | A A | G G | G G | C T | C C | T T | C C | T T | A A | G G | G G | C C | G G | C C | T T | A C | A A |
| Huizao | C T | C T | A A | A G | G T | G G | T T | C C | T T | C T | T T | A A | G G | G T | C T | G G | C C | T T | A A | A A |

**Table 3.** List of 17 synonymous groups, including 49 cultivars, identified by SNP markers in the Chinese jujube collection maintained in Yinchuan, Ningxia, China. The cultivar in bold in the table was retained for subsequent diversity analysis.

| Synon. Group | Cultivar Name | Origin | Synon. Group | Cultivar Name | Origin |
|---|---|---|---|---|---|
| 1 | Cangxiantunzizao | Hebei | 8 | Pozaozhibian1# | Hebei |
| 1 | **Jinsixiaozao** | **Hebei** | 8 | **Dalingzao** | **Shandong** |
| 1 | Laolingxiaozao | Shandong | | | |
| | | | 9 | **Jingudazao** | **Shanxi** |
| 2 | Shulutangzao | Hebei | 9 | Jinchang1# | Ningxia |
| 2 | **Chuanlingzao** | **Hebei** | | | |
| 2 | Bianhesuanzao | Henan | 10 | Yingluozao | Beijing |
| | | | 10 | **Baodeyouzao** | **Shanxi** |
| 3 | **Zanhuangchangzao** | **Hebei** | 10 | Xiangfenyuanzao | Shanxi |
| 3 | Tongxinyuanzao | Ningxia | | | |
| 3 | Dayuanzao | Ningxia | 11 | Goutouzao | Shaanxi |
| 3 | Zhongcaobenzao | Shaanxi | 11 | **Jianjianzao** | **Shanxi** |
| 3 | Wanrongcuizao | Shanxi | | | |
| | | | 12 | Xinzhengjixinzao | Henan |
| 4 | **Shaizao** | **Beijing** | 12 | **Huizao** | **Xinjiang** |
| 4 | Yiwudazao | Zhejiang | | | |
| | | | 13 | **Sumuzao** | **Shanxi** |
| 5 | Minqinxiaozao | Gansu | 13 | Beijingbenzao | Beijing |
| 5 | **Zhongningyuanzao** | **Ningxia** | | | |
| 5 | Yuanzao | Ningxia | 14 | Yunchengpopozao | Shanxi |
| 5 | Yuanzao1# | Ningxia | 14 | **Mopanzao** | **Shanxi** |
| 5 | Yuanzao2# | Ningxia | | | |
| 5 | Yuanzao3# | Ningxia | 15 | **Zaocuiwang** | **Hebei** |
| 5 | Lingwuyuanzao | Ningxia | 15 | Luzao5# | Shandong |
| 5 | Changtanzao | Ningxia | | | |
| | | | 16 | Linfenmugeda | Shanxi |
| 6 | Malianxiaozao | Hebei | 16 | Jiaxianyazao | Shaanxi |
| 6 | Ganweibazao | Shaanxi | 16 | **Muzao** | **Shaanxi** |
| 6 | **Jinzao** | **Shanxi** | | | |
| 6 | Linyibenzao | Shanxi | 17 | **Longzhu2#** | **Ningxia** |
| | | | 17 | Longzhu3# | Ningxia |
| 7 | **Dongzao** | **Shaanxi** | | | |
| 7 | Dongzao2# | Henan | | | |

Based on Shannon's Information Index, a subset of 96 SNP markers was selected (Supplementary Data 2). Every single cultivar could be distinguished by the combined use of these 96 SNPs. The accumulated PID of these 96 SNPs was $6.37 \times 10^{-12}$. Correlation between the full-panel (192 SNPs) and the core-panel (96 SNPs) matrix of genetic distance was highly significant ($r = 0.8075$, $p < 0.01$), as shown by the Mantel Test (Figure 1).

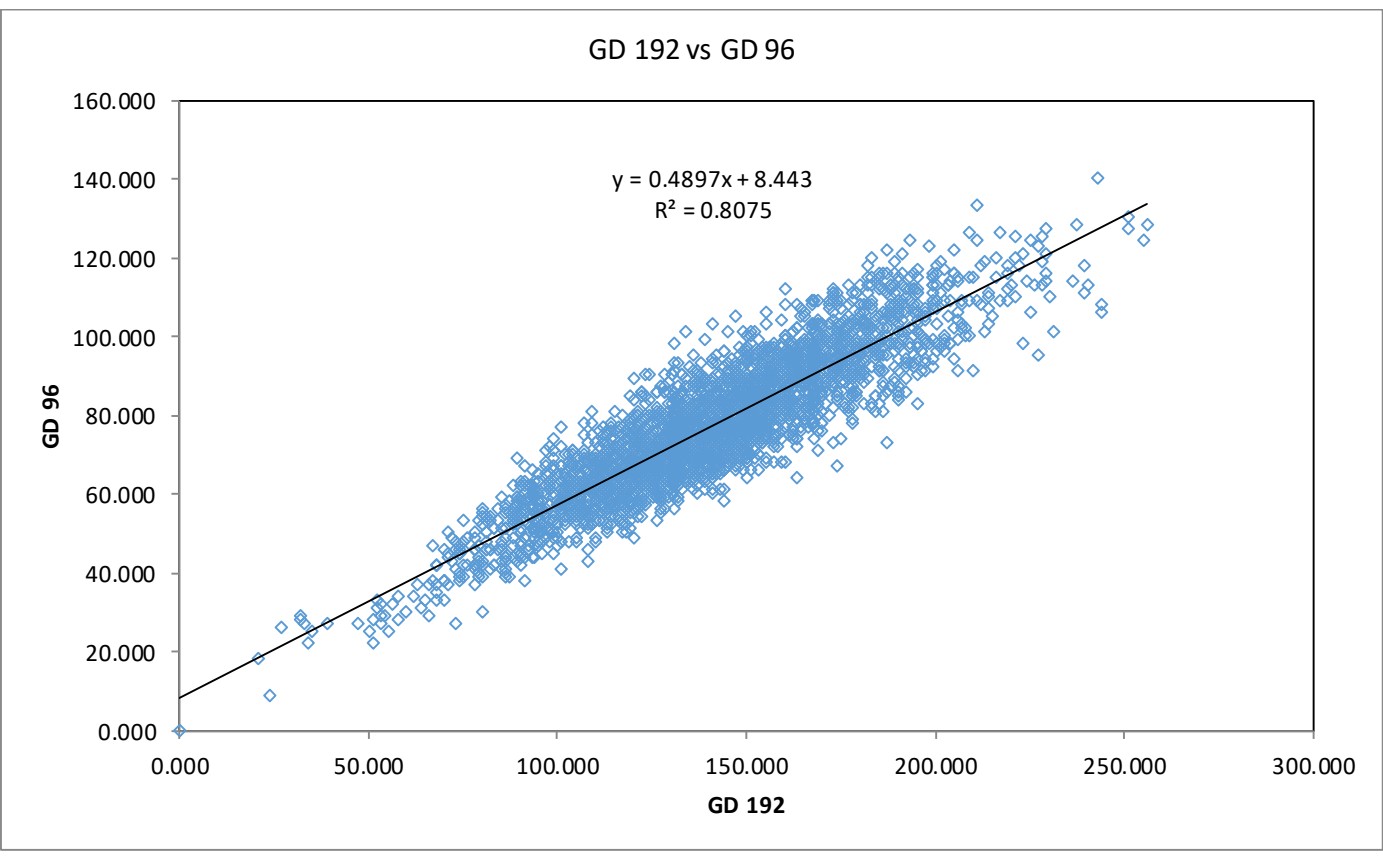

**Figure 1.** Mantel test of correlation between genetic distances generated by the full set of 192 SNPs and the selected core set of 96 SNPs.

### 3.3. Population Stratification

Population stratification of the 79 jujube accessions, based on Δ*K* value computed by STRUCTURE HARVESTER, revealed two clusters (Figure 2) as the most probable number of *K* [22]. At a high assignment coefficient value (Q > 0.80), the first group included 21 core members, whereas the second group included 24. The remaining 34 cultivars were classified as admixed genotypes (Figure 3 and Table 4). The three groups did not show a consistent pattern of geographical origin (i.e., each group included jujube cultivars from different provinces). Nonetheless, in the first group of core members, two-thirds of the cultivars were from Shanxi and Shaanxi, whereas in the second group only 15% of the cultivars were from these two provinces.

Analysis of molecular variance (AMOVA) showed that both the within- and among-group variations were highly significant, accounting for 84% and 16% of the total molecular variance, respectively (Figure 4). Pairwise Fst between the two groups was 0.16, and the result of the permutation test was highly significant (*p* < 0.001), showing a significant genetic differentiation between these two groups.

**Table 4.** Q-values based on Bayesian stratification of 79 Chinese jujube cultivars from Ningxia, China.

| Cultivar | Origin | Q-Value Group 1 | Q-Value Group 2 | Group |
|---|---|---|---|---|
| Beijingpaoapaozao | Beijing | 0.909 | 0.091 | Group 1 |
| Chuanlingzao | Hebei | 0.898 | 0.102 | Group 1 |
| Lingbaoyuanzao | Henan | 0.868 | 0.132 | Group 1 |
| Muzao | Shaanxi | 0.995 | 0.005 | Group 1 |
| Pingshunbenzao | Shaanxi | 0.994 | 0.006 | Group 1 |
| Qiyuexian | Shaanxi | 0.836 | 0.164 | Group 1 |

**Table 4.** *Cont.*

| Cultivar | Origin | Q-Value Group 1 | Q-Value Group 2 | Group |
|---|---|---|---|---|
| Dalingzao | Shandong | 0.991 | 0.009 | Group 1 |
| Linqinzao | Shandong | 0.987 | 0.013 | Group 1 |
| Liaochengyuanlingzao | Shandong | 0.832 | 0.168 | Group 1 |
| Yucituanzao | Shanxi | 0.992 | 0.008 | Group 1 |
| Yuanquzao | Shanxi | 0.989 | 0.011 | Group 1 |
| Jiaochengduanzao | Shanxi | 0.989 | 0.011 | Group 1 |
| Jingudazao | Shanxi | 0.988 | 0.012 | Group 1 |
| Mamazao | Shanxi | 0.983 | 0.017 | Group 1 |
| Linglingzao | Shanxi | 0.978 | 0.022 | Group 1 |
| Yucijiuyueqing | Shanxi | 0.977 | 0.023 | Group 1 |
| Taiyuanyuanzao | Shanxi | 0.961 | 0.039 | Group 1 |
| Mopanzao | Shanxi | 0.960 | 0.040 | Group 1 |
| Jianjianzao | Shanxi | 0.924 | 0.076 | Group 1 |
| Tuanzao | Shanxi | 0.888 | 0.112 | Group 1 |
| Kunmingzao | Yunnan | 0.822 | 0.178 | Group 1 |
| Gansudongzao | Gansu | 0.011 | 0.989 | Group 2 |
| Longzao | Hebei | 0.049 | 0.951 | Group 2 |
| Fengmizao | Hebei | 0.051 | 0.949 | Group 2 |
| Zanhuangchangzao | Hebei | 0.087 | 0.913 | Group 2 |
| Xupusuanyuanzao | Hunan | 0.016 | 0.984 | Group 2 |
| Xupuchengtuozao | Hunan | 0.051 | 0.949 | Group 2 |
| Mifengzao | Hunan | 0.116 | 0.884 | Group 2 |
| Xupushatangzao | Hunan | 0.152 | 0.848 | Group 2 |
| Xupumizao | Hunan | 0.160 | 0.840 | Group 2 |
| Penzao | Jiangsu | 0.011 | 0.989 | Group 2 |
| Zhongningyuanzao | Ningxia | 0.014 | 0.986 | Group 2 |
| Lingwuchangzao | Ningxia | 0.022 | 0.978 | Group 2 |
| Longzhu | Ningxia | 0.022 | 0.978 | Group 2 |
| Longzhu | Ningxia | 0.023 | 0.977 | Group 2 |
| Dongzao | Shaanxi | 0.049 | 0.951 | Group 2 |
| Luzao | Shandong | 0.050 | 0.950 | Group 2 |
| Tengxiantangzao | Shandong | 0.093 | 0.907 | Group 2 |
| Fucuimizao | Shandong | 0.100 | 0.900 | Group 2 |
| Suyuanlingzao | Shandong | 0.163 | 0.837 | Group 2 |
| Jinzao | Shanxi | 0.035 | 0.965 | Group 2 |
| Puchengjinzao | Shanxi | 0.067 | 0.933 | Group 2 |
| Lizao | Shanxi | 0.131 | 0.869 | Group 2 |
| Yiwuezizao | Zhejiang | 0.199 | 0.801 | Group 2 |
| Alaayuancuizao | Xinjiang | 0.217 | 0.783 | Admixture |
| Jingzao60 | Beijing | 0.233 | 0.767 | Admixture |
| Jinsixiaozao | Hebei | 0.243 | 0.757 | Admixture |
| Linfenmizao | Shanxi | 0.262 | 0.738 | Admixture |
| Pingshunjunzao | Shaanxi | 0.266 | 0.734 | Admixture |
| Zunyitianzao | Guizhou | 0.281 | 0.719 | Admixture |
| Baodeyouzao | Shanxi | 0.288 | 0.712 | Admixture |
| Huizao | Xinjiang | 0.300 | 0.700 | Admixture |
| Taigulinglingzao | Shanxi | 0.308 | 0.692 | Admixture |
| Hebei | Hebei | 0.335 | 0.665 | Admixture |
| Dieyazao | Shaanxi | 0.340 | 0.660 | Admixture |
| Hamidazao | Xinjiang | 0.341 | 0.659 | Admixture |
| Suanzao | Ningxia | 0.363 | 0.637 | Admixture |
| Henanlongzao | Henan | 0.406 | 0.594 | Admixture |
| Zaoqiuhong | Ningxia | 0.429 | 0.571 | Admixture |
| Lingwusuanzao | Ningxia | 0.444 | 0.556 | Admixture |
| Xianxianmatouzao | Hebei | 0.473 | 0.527 | Admixture |
| Zaocuiwang | Hebei | 0.490 | 0.510 | Admixture |
| Luzao | Shandong | 0.492 | 0.508 | Admixture |

**Table 4.** *Cont.*

| Cultivar | Origin | Q-Value Group 1 | Q-Value Group 2 | Group |
|---|---|---|---|---|
| Dalixiaodundunzao | Shaanxi | 0.526 | 0.474 | Admixture |
| Yiwumianxuzao | Zhejiang | 0.550 | 0.450 | Admixture |
| Sumuzao | Shanxi | 0.558 | 0.442 | Admixture |
| Ruchengzao | Hunan | 0.562 | 0.438 | Admixture |
| Dasuanzao | Ningxia | 0.589 | 0.411 | Admixture |
| Shaizao | Beijing | 0.619 | 0.381 | Admixture |
| Linyitiansuanzao | Shanxi | 0.625 | 0.375 | Admixture |
| Yunchengjinzao | Shanxi | 0.644 | 0.356 | Admixture |
| Lengsizao | Jiangsu | 0.649 | 0.351 | Admixture |
| Beibeixiaozao | Chongqing | 0.680 | 0.320 | Admixture |
| Jinmangguozao | Henan | 0.708 | 0.292 | Admixture |
| Yongjijidanzao | Shanxi | 0.714 | 0.286 | Admixture |
| Jiuyuehan | Shanxi | 0.723 | 0.277 | Admixture |
| Fuyangmutouzao | Anhui | 0.728 | 0.272 | Admixture |
| Tangtouzao | Hunan | 0.768 | 0.232 | Admixture |
| Xipushibingzao | Hunan | 0.793 | 0.207 | Admixture |

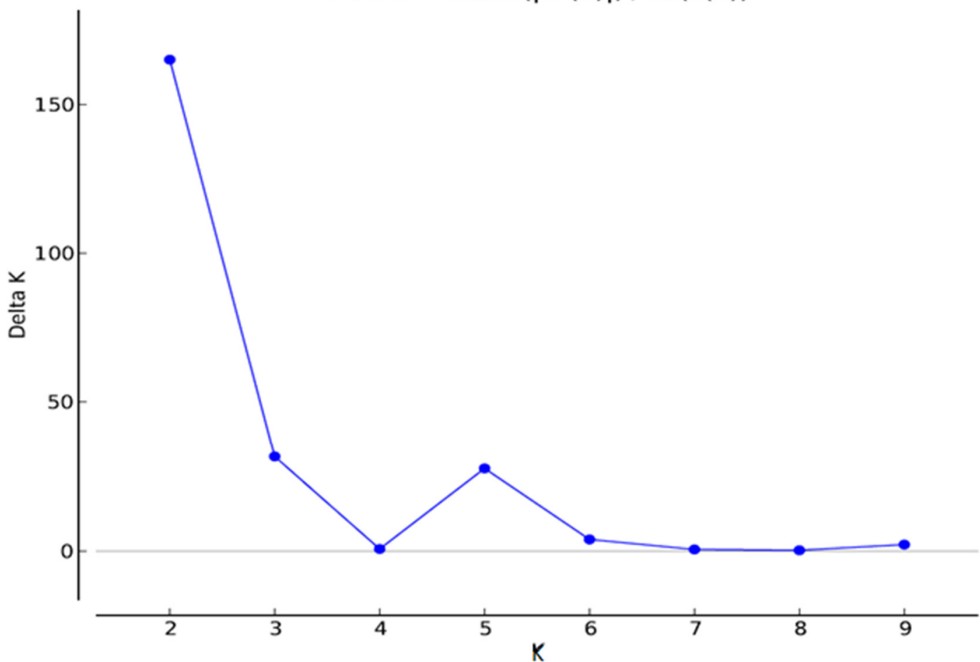

**Figure 2.** Plot of ΔK (filled circles, solid line) calculated as the mean of the second-order rate of change in likelihood of K divided by the standard deviation of the likelihood of K, m(|L''(K)|/s[L(K)].

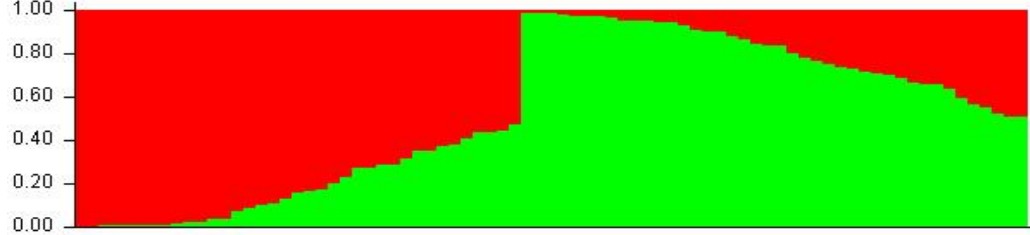

**Figure 3.** Inferred clusters in the 79 jujube cultivars (and synonymous groups) using STRUCTURE in the overall analyzed jujube accessions. Each vertical line represents one individual multilocus genotype. Individuals with multiple colors have admixed genotypes from multiple clusters. Each color represents the most likely ancestry of the cluster from which the genotype or partial genotype was derived. Clusters of individuals are represented by colors.

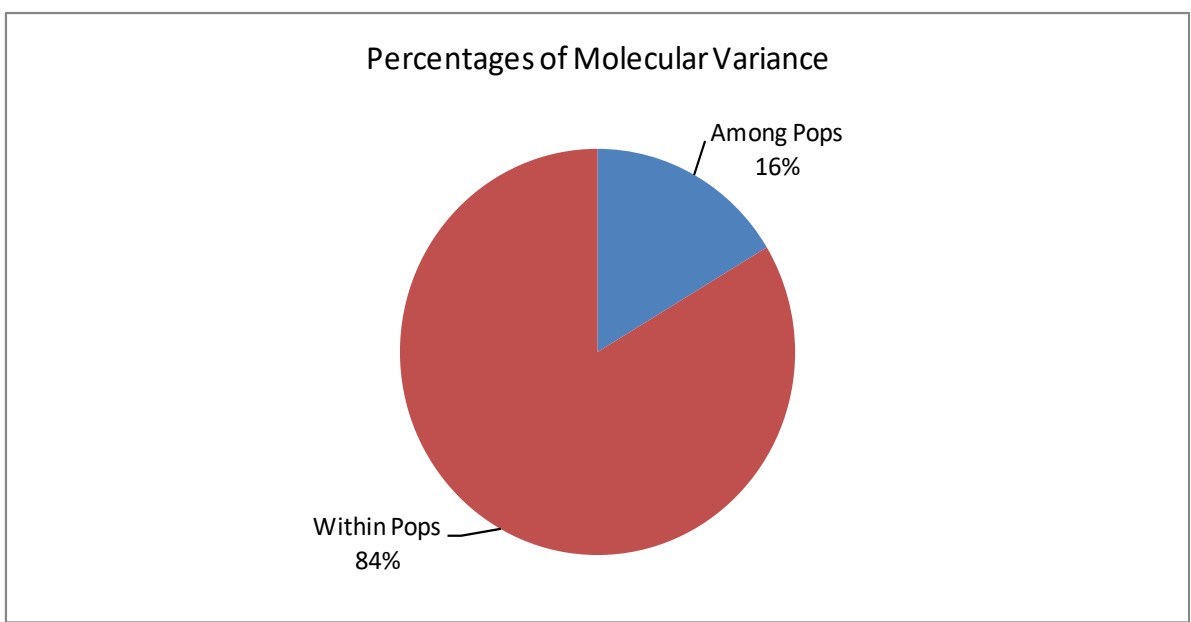

**Figure 4.** Analysis of Molecular Variance of core members in the two jujube cultivar groups assigned by Structure program.

*3.4. PCoA and Clustering Analysis*

Genetic relationships among the analyzed jujube accessions are presented in the principal coordinates analysis (PCoA) plots (Figure 5). The two core member groups assigned by the Bayesian clustering analysis were clearly distinguished without overlapping, showing the different genetic background between these two groups of cultivars. However, the geographical pattern was not clearly reflected in the PCoA.

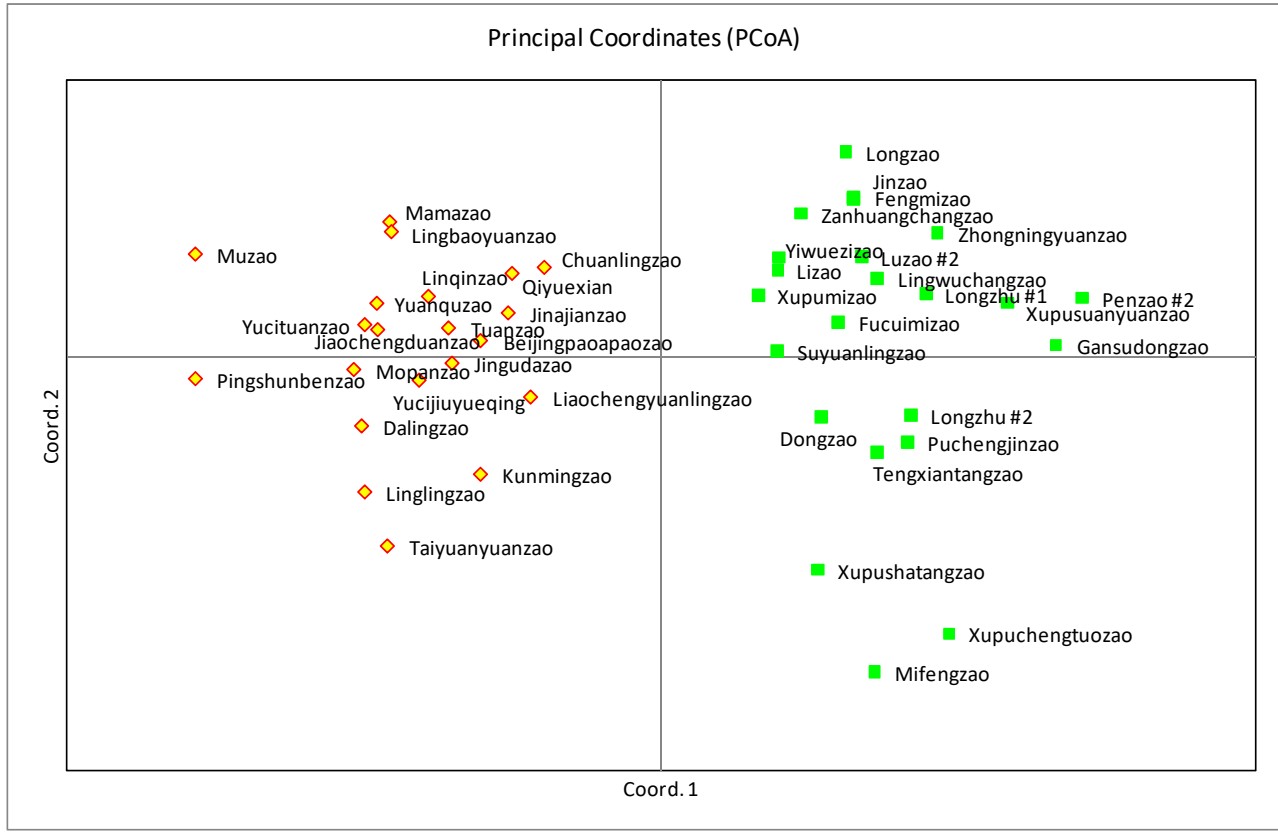

**Figure 5.** Principle Coordinate Analysis of the two distinctive jujube cultivar groups in 43 cultivars.

The NJ tree revealed additional insight that is complementary to those presented by PCoA and Bayesian stratification (Figure 6). The NJ tree classified the jujube cultivars into 20 small sub-clusters, which were deeply separated. However, each of these 20 subclusters comprised 1–13 closely related cultivars. Some of these sub-clusters, such as Huizao, Ruchengzao, Lizao, Longzhu 1, etc. reflected specific geographical origins.

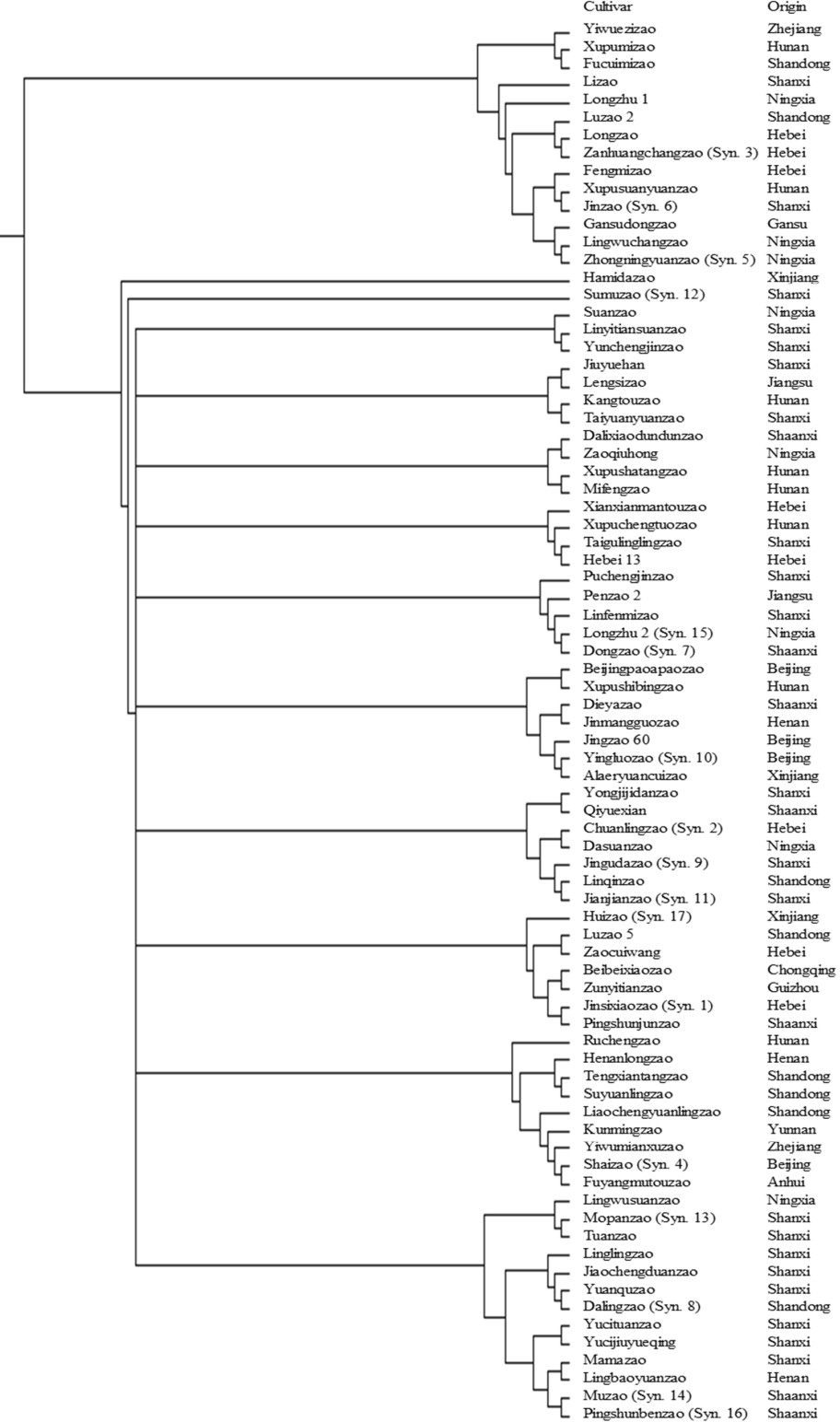

**Figure 6.** Neighbor-joining tree depicting the relationship among 79 jujube cultivars (and synonymous groups) from Ningxia jujube germplasm collection, Yinchuan, Ningxia. Identification of accessions corresponds to samples listed in Table 2.

### 3.5. Parentage Analysis

Among the five cultivars that were indicated as true seedlings, four of them were assigned a paternal (or maternal) parent (>95% confidence level) by matching with the records (Table 5). The only exception is 'Zaoqiuhong', whose maternal parent was supposed to be 'Dalingzao' from Shandong. The result further clarified that cv. 'Longzhu 1' and 'Longzhu 2', which were thought to be siblings from the same parents, were not related and had different parents. 'Longzhu 1' was a progeny of 'Lizao', whereas 'Longzhu 2' was a progeny of 'Dongzao'. The result of parent–offspring assignment is also highly compatible with the cluster analysis (Figure 6), where the identified parent–offspring pairs were all grouped closely in the same sub-clusters.

**Table 5.** Testing recorded parent—offspring relationship in jujube cultivars using 192 SNP markers.

| Cultivar | Recorded Parent | Identified Parent | Pair Top LOD |
|---|---|---|---|
| Jing 60 | Uknown landraces from Beijing | Yingluozao (Syn. 10) | 1.91 |
| Zaocuiwang | Jinsixiaozao | Jinsixiaozao (Syn. 1) | 4.91 |
| Qiyuexian | Known landraces | Yongjijidanzao | 3.66 |
| Longzhu 1 | Lizao and/or Dongzao | Lizao | 1.41 |
| Longzhu 2 | Lizao and/or Dongzao | Dongzao (Syn. 7) | 3.01 |
| Zaoqiuhong | Dalingzao | Dalixiaodundunzao | 7.18 |
| Luzao 2 | Liuyuexian | Zanhuangchangzao (Syn. 3) | 2.52 |

## 4. Discussion

### 4.1. Development of SNP Markers through Data Mining

Despite great progress in genomics research on jujube, availability of advanced molecular tools to support germplasm management has been scarce. Developing SNP markers using available sequences could fill the gap between genomic research and downstream applications by jujube breeders and genebank curators. In the present study, we developed 32,249 putative SNPs based on SRA sequences of jujube in a public database and used them to genotype a diverse panel of 114 jujube cultivars. We obtained a success rate of approximately 80% for marker validation, which demonstrated that this approach is effective and can thus serve as a shortcut for large-scale SNP development.

### 4.2. Jujube Cultivar Identification Using SNP Markers

Reliable identification of jujube cultivars is invaluable for management of jujube genetic resources, propagation of planting materials, and breeding for new cultivars with desirable agronomic traits and quality attributes. In the present study, it has been demonstrated that the SNP marker fingerprinting was effective for the assessment of genetic identity of jujube germplasm. As shown in the present study, results from multiple clones of the same cultivar showed 100% concordance, demonstrating that the nanofluidic array system is a reliable platform for generating jujube DNA fingerprints with high accuracy.

The present results revealed a high rate of genetic redundancy in the tested jujube collection. This result is consistent with the result of Xu et al. (2016), who reported that 47% of the analyzed germplasm accessions had at least one duplicated accession. This high rate of synonymous mislabeling can be explained by the fact of germplasm exchange. Jujube has a long cultivation history in China. Elite cultivars were introduced to different regions and the long-term interregional cultivar exchange has resulted in extensive duplications in germplasm collections. Some of the identified duplicates are well-documented synonymous cultivars. For example, 'Jinsixiaozao' is a popular cultivar widely distributed in the provinces of the lower Yellow River valley, such as Shandong, Henan, Hebei, and Beijing. As shown in the present study, the same cultivar was labeled differently in different regions (e.g., 'Jinsixiaozao', 'Laolingxiaozao', 'Cangxiantunzizao', and 'Puyangxiaozao'), which caused duplications in ex situ genebanks. The same patterns were found in elite cultivars 'Zanhuangchangzao', 'Zhongningyuanzao', and 'Minqinxiaozao'. Identification of these

synonymous groups will significantly improve the accuracy and efficiency in the exchange, conservation, and use of jujube germplasm.

However, caution needs to be taken regarding the interpretation of cultivars with the same SNP profiles. This is because somatic mutations are commonly reported in jujube and can modify many phenotypic traits such as fruit skin color, flesh color, growth habit and fruit quality attributes [3]. These somatic mutations have been the major source of variation exploited for the selection of new cultivars. For example, between 2007 and 2014, there were 11 newly released cultivars in China that were selections based on somatic mutations [33]. This challenge also existed in fingerprinting projects dealing with other vegetative propagated crops, such as pineapple [10] and banana [34]. For these types of duplicates, more comprehensive genomic approaches, such as genome resequencing, would be needed for the detection of somatic mutations and copy number alterations in corresponding genes or alleles. For this reason, the reduction of genetic redundancy in jujube genebank should not be based on DNA fingerprints only. Characterization of phenotypic traits of the synonymous group members is still essential to complement DNA fingerprinting for genotype identification.

In addition, genotyping of the jujube collection in Ningxia alone is not sufficient to fully correct the mislabeling in this collection. This is because most of the jujube accessions in the Ningxia repository were introduced from various jujube collections in other provinces in China. These germplasms are not necessarily authentic. Therefore, to correct the mislabeling in these introduced cultivars, the reference profiles of the original trees in the source genebanks need to be established using the same set of SNP markers. These reference SNP profiles then can be compiled and deposited in a jujube germplasm database, which should be publicly accessible, in order to make comparisons between reference standard and any tested cultivars or clones.

### 4.3. Parentage Verification for Improved Jujube Cultivars

In addition to accurate cultivar identification, accurate parentage and pedigree information is also imperative for jujube cultivar registration and protection of the breeder's rights as well as for efficient use of germplasm in breeding programs. Although most jujube cultivars currently used in production are landraces, improved cultivars and breeders' selections are being released at an accelerated pace [9]. However, the recorded parentage has not always been clear for the released cultivars. Moreover, the recorded parents could be a mislabeled accession. The present study evaluated the efficacy of using the developed SNP panel for parentage verification. Among the five cultivars that have known parental cultivars or parentage background, four were proven to have the correct parents–offspring relationship, and one was found to be misreported.

The discrepancy between breeding records and observed SNP profiles was well illustrated by the example of 'Longzhu 2' and 'Longzhu 3'. These two varieties were recorded as siblings selected from the same progeny population of 'Dongzao' × 'Lizao'. However, these two were found to be duplicates in the present study, suggesting the possibility that the breeding record might be wrong. Nonetheless, more samples of 'Longzhu2', and 'Longzhu3', preferably from original sources, need to be examined to confirm the observation. The results demonstrate the usefulness of using these SNP markers to support jujube cultivar registration. Given that hybrid verification is of critical importance in jujube breeding because of self-compatibility in some germplasm accessions [9,32], these SNP markers could also be used by jujube breeders to effectively manage breeding lines based on marker-based parentage and family pedigree.

### 4.4. The Core Set of SNP Markers for Universal Jujube Cultivar Identification

Various molecular markers have been applied on jujube cultivar identification. However, the key challenge is to have a standard set of markers that can allow cross-laboratory data comparison. Despite the high polymorphism of SSR markers, it is difficult to compare and combine SSR fingerprints generated by different laboratories or genotyping platforms

(e.g., ABI, SEQ, or other gel box electrophoresis). Additional challenges include the low accuracy, low efficiency, and high cost. Therefore, a small core set of SNP markers is needed for various downstream applications in the value chain of jujube. Data generated by this small set of SNPs can be easily compared with each other, regardless of the genotyping platform used.

The present study selected 96 high quality SNPs (out of the 192 SNPs reported here), which formed a jujube genotyping kit. This subset of SNP markers was filtered to remove markers that show a high level of linkage disequilibrium (LD) and have a high polymorphism informative content (PIC). The accumulated PID value demonstrated that this panel has sufficient statistical power for accurate cultivar identification of jujube cultivars. The Mantel test showed a high correlation (r = 0.91) between these 96 SNPs and the full panel of 192 SNPs. The generated SNP profiles can be converted into a simple bar code and be used in many other downstream applications, such as nursery accreditation, cultivar registration, and the authentication of geographically referenced jujube products.

### 4.5. Genetic Relationships among the Different Germplasm Groups

Bayesian stratification (Figure 3) showed that the 79 unique jujube cultivars (and synonymous groups) could be grouped into two different clusters. The partitioning result did not show a consistent pattern of geographical origin. The Fst value between the two germplasm groups was 0.16, demonstrating a substantial interpopulation differentiation and therefore supporting the hypothesis of significant regional differentiation of jujube germplasm [1,3].

Since the Evanno Delta K graph also showed secondary peaks at K = 3 and K = 5, we included the corresponding partitioning results in the Supplemental Data 5. However, it is worth noting that at K = 3 and K = 5, a much larger proportion of the cultivars were classified as admixture. This was likely due to the relatively small sample size used in the present study. Indeed, out of the 800 or so existing germplasm accessions, only a small fraction was included in the present study. Full-scale sampling of the cultivated jujube gene pool, together with established reference standards, will be needed to correctly partition the jujube varieties into appropriate genetic clusters.

The NJ Tree revealed complementary insight about the relationships among the 79 cultivars. The 19 small sub-clusters were deeply separated in the NJ tree (Figure 6), which suggests that there was a lack of crosses and recombination among these sub-clusters. However, each sub-cluster comprised several (up to 13) closely related cultivars, and some of them were exclusively from the same region. This observation indicates that these closely related cultivars may share a common ancestry or parentage. This type of clustering pattern suggests that the large number of jujube cultivars (>800) in China could have been derived from a much smaller number of progenitors that have not been crossed with each other extensively, either due to geographical separation or reproductive barrier (e.g., cross-incompatibility and self-fertilization).

This interesting pattern of genetic structure in jujube germplasm suggests that there is great potential to explore heterosis between the germplasm cluster and sub-clusters. From the perspective of long-term germplasm conservation and genebank management, the present results also suggest that a much smaller collection can be sampled to represent most of the genetic diversity existing in the large number of jujube cultivars. In this way, more resources could be allocated to conserving other related taxa and ensure that maximum genetic diversity in the primary gene pool of jujube is conserved.

In conclusion, we conducted a study to develop a large number of SNP markers for jujube germplasm management and genetic improvement. We validate a small set and applied them for fingerprinting the jujube germplasm collection in Ningxia, China using a nanofluidic array method. This approach enabled us to generate high-quality SNP profiles for accurate identification of jujube cultivars. This tool is highly useful for the management of jujube genetic resources, which will also lead to more efficient selection of parental clones for jujube breeding. Furthermore, these SNP markers can be used to protect

intellectual property rights of breeders, monitor clone purity of planting materials, and for the authentication of premium jujube products. Our result also generated significant insight regarding the classification of jujube cultivars. For the identified synonymous groups, morphological characterization is underway to identify any somaclonal mutations that may have occurred in these synonymous groups. Genome resequencing will be applied to gain a comprehensive understanding of the genetic basis for mutation-based changes in important agronomic traits. This SNP-based genotyping approach will be highly useful in many other areas of the jujube industry.

**Supplementary Materials:** The following are available online at https://www.mdpi.com/article/10.3390/agronomy11112303/s1, Supplementary Data 1. Full list of 32,249 putative SNP markers and associated information identified using data mining approach. Supplementary Data 2. 192 SNPs and their flanking sequences retained in data analysis of present study. The top 96 SNPs were selected based on their high value of Shannon's Information Index. Supplementary Data 3. SNP based DNA fingerprints generated by the 192 SNP markers for all 114 analyzed Chinese jujube cultivars. Supplementary Data 4. Summary statistics, including information index, observed heterozygosity, and gene diversity of 192 SNP markers selected for Chinese jujube cultivar identification. Supplementary Data 5. Inferred clusters in the 79 jujube cultivars (and synonymous groups) using STRUCTURE in the overall analyzed jujube accessions at K = 3 and K = 5. Each vertical line represents one individual multilocus genotype. Individuals with multiple colors have admixed genotypes from multiple clusters. Each color represents the most likely ancestry of the cluster from which the genotype or partial genotype was derived. Clusters of individuals are represented by colors.

**Author Contributions:** L.S., D.Z. and B.C. designed the experiment, analyzed data, and wrote the manuscript. Y.Z. conducted the experiment and interpreted the analytical result. L.W.M. and B.C. revised and edited the manuscript. All authors have read and agreed to the published version of the manuscript.

**Funding:** This research received no external funding.

**Institutional Review Board Statement:** Not applicable.

**Informed Consent Statement:** Not applicable.

**Data Availability Statement:** All relevant data are within the paper and its Supplementary Information Files.

**Acknowledgments:** We would like to give special thanks to Huawei Tan for data mining and identification of SNP markers, Stephen Pinney of USDA-ARS for SNP genotyping of the jujube samples, and Sue Mischke for editing the manuscript. This work was partially supported by the Project of the Key Research and Development Projects in Ningxia (2018BFH03015) and Natural Science Foundation of Ningxia (2021AAC03110).

**Conflicts of Interest:** The authors declare no conflict of interest.

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
