# Peer review of "Mining Single Nucleotide Polymorphism (SNP) Markers for Accurate Genotype Identification and Diversity Analysis of Chinese Jujube (Ziziphus jujuba Mill.) Germplasm"

_agronomy, doi:10.3390/agronomy11112303_

Round 1

Reviewer 1 Report

The manuscript is very well written. The identification of a core set of SNP markers as fingerprints for identifying germplasm collections will  be very useful in genebank management and for breeding programs. I suggest some minor edits. Comments can be found in the attached manuscript. 

Author Response

Dear reviewer,

Thanks for your reports, comments and suggestions on our manuscript (agronomy-1432805). We really appreciate it very much. We thank you for taking time and effort that they have put into reviewing the previous version of the manuscript. Your suggestions have enabled us to improve our manuscript. Accordingly, we have uploaded a copy of the original manuscript with all the changes highlighted by using the track changes mode in MS Word. Appended is our point-by-point response to your comments. Thanks again.

Sincerely,

Bing Cao & Lihua Song

Point-by-point response:

(1) Line 36, page1, All references should be "et al." instead of "el at." Please change throughout the manuscript.  We have modified throughout the text according to the comment.

(2) Line 97, page2, If there are only two authors, both should be listed. This should be Li & Durbin, 2009.   We have modified throughout the text according to the comment. (line 117,page 3; line 152, line 161, page 5; line 206, page 6; )

(3) Line 204, page 6, Delete “based on”. We have modified.

(4) Line 256, page 7, Within each synonymous group, identify which cultivar was retained for subsequent diversity analysis. Perhaps that cultivar can be in bold in the table with an explanation in the table caption. We have modified according to the comment.

(5) Line 298-299, page 9, Figure 4B is really not needed and can be removed. We removed the fig. 4B.

(6) Line 317, page 14, This should be Table 2. We have modified according to the comment.

(7) Line 332, page 14, This cultivar belongs to Syn. Group 10 and not 6. We have modified according to the comment.

(8) Line376, page 15, This citation is not in the Reference List. We have modified according to the comment.

(9) Line405, page 15, Check spelling - should this be Longzhu 3? We have checked and modified. (10) Line496, page 17, D.Z., B.C. We have modified according to the comment.

(11) Line528-530, page 18,  Not cited in text. We have modified

(10) Line531-532, page 18,Check format - The year 2014 should be at the end. We have modified.

Reviewer 2 Report

The article is well written and it have an interesting topic  about cultivar identification and genetic diversity in jujube. Some minor correction its needed that was indicated as highlighted and comments in attachment file.

Author Response

Dear reviewer,

Thanks for your reports, comments and suggestions on our manuscript (agronomy-1432805). We really appreciate it very much. We thank you for taking time and effort that they have put into reviewing the previous version of the manuscript. Your suggestions have enabled us to improve our manuscript. Accordingly, we have uploaded a copy of the original manuscript with all the changes highlighted by using the track changes mode in MS Word. Appended is our point-by-point response to your comments. Thanks again.

Sincerely,

Bing Cao & Lihua Song

Point-by-point response:

(1) Line 50, page2, references should be "et al." We have modified throughout the text according to the comment.

(2) Line 129, page3, there is extra space. We have modified throughout the text according to the comment. ( line 201, page 5.)

(3) Line 230, page6, missing a space. We have modified.

(4) Line 269, page8, the “R?” and wrong display for ordinate title. We have checked, and the display at original fig.1 for the manuscript is normal, this must be an editing and printing problem. Same situation as line 297, page 9; line 310, page 12;

(5) Line 274, page8, revealed two or three clusters? We have checked, and should be revealed two clusters.

(6) Line 311, page12, the figure 5 title should be added “in 43 cultivar”. We have modified according to the comment.

(7) Line 318, page14, The NJ tree have two main clusters, 13 sub cluster. How 19 sub cluster are assigned? The genetic distance index (mark) is not seen in the figure. We have checked and added the classified genetic distance index (mark), and modified as 20 sub clusters.

(8) Line 443, page16, The Data 5 was not in supplementary files. We have checked and added the data 5 in supplementary files.

Reviewer 3 Report

Comments to the authors

The focus of the manuscript ‘Mining Single Nucleotide Polymorphism (SNP) markers for

accurate genotype identification and diversity analysis of Chinese jujube (Ziziphus jujuba Mill.) germplasm’ was to develop a set of highly polymorphic SNP markers which can be used in future as reference marker set for genotyping and parentage analysis of jujube cultivars. The efficiency of the SNP markers was demonstrated by their application in a germplasm collection of Chinese jujube, where genetically identical cultivars and diversity parameter were successfully determined.

The manuscript is nicely written and presents valuable findings about the usefulness of SNP markers in jujube gene bank and breeding management.

Please see the comments in the manuscript.

Author Response

Dear reviewer,

Thanks for your reports, comments and suggestions on our manuscript (agronomy-1432805). We really appreciate it very much. We thank you for taking time and effort that they have put into reviewing the previous version of the manuscript. Your suggestions have enabled us to improve our manuscript. Accordingly, we have uploaded a copy of the original manuscript with all the changes highlighted by using the track changes mode in MS Word. Appended to this letter is our point-by-point response to your comments. Thanks again.

Sincerely,

Bing Cao & Lihua Song

Reponse to the Reviewer‘s’ comments:

(1) Line 20-21, page1, “stratification” is correct.  You mean that the core members showed a high genetic differentiation? We revised the sentence as “and the core members of the two groups showed a significant genetic differentiation”

(2) Line 36, page1, please correct el at in et al (every reference). We have modified throughout the text according to the comment.

(3) Line 41, page1, there is an extra Superscript. We have modified.

(4) Line 47, page2, “Presently it ranks 7thamong fruit tree crops”, we have modified as “Presently it ranks seventh among fruit tree crops ”.

(5) Line 52, page2, there missing a space after word “China”. We have modified.

(6) Line 66, page2, Check format of reference. We have checked and modified.

(7) Line 74, page2, “Lack of correctly identified jujube germplasm was also found to have severely limited the development of improved jujube cultivars.”, This statement is not clear for me. Please can you give a short explanation? We have added a clarification. The sentence was modified as “Lacking of correctly identified jujube germplasm has hindered the use of true-to-type parental lines thus severely limited the development of improved jujube cultivars.”

(8) Line 79, page2, please rewrite this sentence, “Compared to SSR markers, SNP analysis doesn’t require DNA separation by size.” We have modified as “Compared to SSR markers, SNP analysis doesn’t require DNA separation by fragment length.”

(9) Line 88, page2, please check the text for format errors. We have checked and modified throughout the text according to the comment.

(10) Line 113, page3, please add also the NCBI accession code. We have modified and added.

(11)Line 117, page3, we modified “the filter” as “exclude”.

(12)Line 150, page5, How did you identify identical genotypes? Using also Gen Alex? Please specify.  We added: The procedure of mulit-locus matches, as implemented in the program GenAlEx 6.5 (Peakall & Smouse, 2006), was used for computation.

(13)Line 154, page5, “sibling individuals”,The PID assumes that individuals to be compared are unrelated. Correction was made. We meant to say PID-sib in the text.  The sentence was revised as: The probability of identity (PID among siblings (PID-SIB), which is the probability that two sibling individuals drawn at random…

(14)Line 154, page5,  Please change 'population' to 'collection'. We have made revision.  

(15)Line 156, page5, necessary number and most useful loci for? We have changed “most conservative” as “minimum essential”.

(16)Line 165, page5, we have modified, using the 'number of genetic clusters' instead of 'number of subpopulations'.

(17-18) Line 186 - 187, page5, the sentence was revised as “seven cultivars with known parents (per literature records), were selected for parentage analysis (Table 1). It’s known that these seven cultivars were selected from true seedlings (in contrast to clonal selection) but their parentage was either partially known or unknown.

(19)Line 221, page6,  The sentence was changed as “SNP sites having an adjacent SNP site either 80 bp upstream or 80 bp downstream were eliminated.”

(20)Line 243, page6   Correction made.

(21)Line 246, pageGood point.  We didn’t carry out phenotypic characterization in the present study. However we acknowledged the necessity in discussion:  Line 368: “However, caution needs to be taken regarding the interpretation of cultivars with same SNP profiles. This is because somatic mutations are commonly reported in jujube and can modify many phenotypic traits such as fruit skin color, flesh color, growth habit and fruit quality attributes (Li, 2015)……

(22)Line 251, page7  Correction made. 

(23)Line 257, page7 Good point.  However, we don’t have the cultivars from the original sources yet.  Therefore, at the present stage, we can only list the synonymous group, without knowing exactly which one is the original cultivar.  This limitation was acknowledged in the discussion.  

(24)Line 267, page8, In the figure 1 the correlation coefficient is R = 0.8075 instead of r=0.899. Please check. We have checked and modified.

(25)Line 268, page8, We have checked, and the display at original fig.1 for the manuscript is normal, this must be an editing and printing problem.

(26)Line 272, page8,  please check this term, 'stratification'.  “stratification” is correct.

(27)Line 297, page9, we improve the quality of figure 4, and removed fig. 4B, changed 'pops' into 'group'.

(28) Line 302, page10, we add the Q-value group 1 and Q-value group 2 in the table 4 column.

(29) Line 302, page10, about the table 4. At what threshold value is the cultivar indicated as admixture? In the text you have indicated only 2 groups, here you have 3 groups (1,2 and admixture). I recommend to indicate 'admixture (group 1)' or 'admixture (group 2)'.  The threshold value was set at Q=0.80.  Cultivars with Q-value lower than 0.80 were classified as admixture. This was stated in page 8, line 273: “Population stratification of the 79 jujube accessions, based on ΔK value computed by STRUCTURE HARVESTER, revealed two clusters as the most probable number of K”. 

(30) Line 319, page14, “Some of these sub-clusters reflected specific geographical origins. ”, Which one? We have checked and modified as “Some of these sub-clusters, such as Huizao, Ruchengzao, Lizao, Longzhu 1, etc. reflected specific geographical origins.”

(31) Line 322-323, page14, we rewrite this sentence as “Among the five cultivars that were indicated as true seedlings, four of them were assigned paternal (or maternal) parent (>95% confidence level) matching with the records (Table 5).”

(32) Line 405, page15, we changed “Lingzhu” into “Longzhu”.
